# Influence of Common Reducing Agents on Technological Parameters of Dry-Fermented Sausages with Low Fat Content

**DOI:** 10.3390/foods11172606

**Published:** 2022-08-27

**Authors:** Montserrat Vioque-Amor, Rafael Gómez-Díaz, Ignacio Clemente-López, Maite Sánchez-Giraldo, Carmen Avilés-Ramírez

**Affiliations:** 1Research Group AGR-120, Department of Food Science and Technology, University de Cordoba, Ctra. Madrid-Cadiz km 396, 14071 Cordoba, Spain; 2DOSCADESA 2000 S. L., Molina de Segura, 30500 Murcia, Spain

**Keywords:** ascorbic acid, starter culture, sodium ascorbate, glucono-delta-lactone

## Abstract

The production of dry-fermented sausages currently presents several challenges to be addressed: nutrition, health, sensory traits and technology are among the main issues that concern consumers and the meat industry. The objective of this study was to evaluate the effect of different reducing agents commonly used in the manufacture of dry-fermented sausages (salchichon type) with low fat content on the technological characteristics of the product. Four different reducing agents (ascorbic acid, a starter culture, sodium ascorbate and glucono-delta-lactone) were added to the meat batter to assess their impact on the physico-chemical traits, instrumental color, residual nitrates and nitrites and lipid oxidation of this meat product. High nitrate values were observed during both ripening and storage periods. All batches presented lipid oxidation values below the threshold that indicates meat rancidity. Adjustments of the original lean:fat proportion must be carried out on this low fat content sausage to obtain profitable results. Starter culture and sodium ascorbate have shown the best reducing and antioxidant activities among the additives studied. Once we have the technical design of the product, the next step will be oriented to identify the sensory characteristics in order to find a gap in the market.

## 1. Introduction

Dry-fermented sausages are well-appreciated food specialties linked in many cases to cultural heritage, which many consumers are determined to not give up eating. Aroma, flavor, taste and texture are among the main sensory characteristics that drive consumers towards purchasing and consuming convenience food products [1]. However, it has been documented that meat product consumption is related to the prevalence of a wide range of emerging diseases such as obesity, cancer, heart-related diseases and other disorders [2]. This is one of the reasons for growth in the number of consumers concerned about the healthiness of food in the last few decades. In this context, the reduction in fat content and its substitution by different alternatives has been deeply studied in dry-fermented meat products such as Spanish salchichón [3] and chorizo [4] in recent years. Although a reduction in fat content would improve the nutritional quality of these sausages, it is known that sausages with low fat content are less appreciated by consumers.

The production of dry-fermented sausages presents more challenges: unclear evidence has been presented that connects nitrate/nitrite intake and the formation of nitrosamines; therefore, the use of nitrate/nitrite is currently quite polemic within the scientific community and health-related authorities [5]. Although in recent years the use of vegetal extracts has been widely studied as an alternative to regulate nitrosamine formation, they present some disadvantages in relation to conventional reductants being mainly linked to sensory attributes [6]. Traditional reducing agents that also enhance the inhibitory effect of nitrite towards *C. botulinum*, such as ascorbic acid or sodium ascorbate [7], or towards *Listeria* spp., such as glucono-delta-lactone [8], are commonly used in small concentrations in processed foods. The addition of bioprotective lactic acid bacteria cultures is also an alternative that guarantees the exhaustive control of the manufacturing process and nitrosamine formation [9]. Apart from the amount and the nature of reductants present, many factors affect the nitrite reactions, including meat pH, temperature or time [10]. The control of initial nitrate/nitrite content and residual nitrite in meat products is important and, in the food industry, the use of these additives is strictly regulated [11]; however, their reduction or substitution is still a challenging task [9]. In fact, a working document with a revision of the current limits of nitrate/nitrite levels has been drafted by the European Food Safety Authority for cheese, fish and fishery products and meat products [12].

Modern lifestyles have also led the meat industry to offer meat products which are ready for consumption that must be increasingly attractive, clean and healthy. The manufacture of these products uses and promotes new processing methods and technologies to meet new consumer expectations [13].

Thus, the reformulation of meat products in order to adjust the values of critical substances is becoming ever more necessary. For this purpose, it is essential to figure out the behavior and properties of each ingredient in detail during the manufacture of meat products. Above all, this is most important in products subjected to a ripening stage in which biological processes are more unpredictable (such as fermentations).

The development of new formulations to achieve a reduction in fat level is a common strategy followed by the industry to improve the nutritional value of traditional high-fat foods such as dry-fermented sausages [14]. The use of reducing agents (known as cure accelerators) is also widespread in these products. Its aim is to diminish health risks by reducing the amount of nitrate used when it is added during the manufacturing process [15]. Additionally, the application of commercial ready-to-use nitrate-reducing starter cultures is currently used as an alternative for curing processes [16]. All these approaches seek to improve the nutritional quality of fermented sausages. To our knowledge, there are not many studies that contemplate both of these issues in dry-fermented meat products: fat reduction in the formula together with the use of agents that reduce nitrifying salts.

Therefore, the aim of the present work was to evaluate the effect of ascorbic acid, a starter culture, sodium ascorbate and glucono-delta-lactone as common reducing agents used in the manufacture of dry-fermented sausages (salchichón type) with low fat content on the technological characteristics of the product. With this study, we analyzed in detail the evolution of these parameters during the ripening process and the storage period in order to compare the effectiveness of different additives and to adjust the composition of such a dry-fermented product.

## 2. Materials and Methods

### 2.1. Experimental Design and Preparation of Sausages

To prepare a common batter for manufacturing the sausages, the following ingredients were mixed: minced lean pork (63.8%), minced pork belly (31.75%), NaCl (2.1%), dextrin (1.0%), dextrose (0.6%), trisodium citrate (0.1%), potassium nitrate (0.25%), sodium acid pyrophosphate (0.15%) and tripolyphosphate (0.25%). A negative control (C-) with no reducing agents and four different batters corresponding to the four experimental formulas with varied reducing agents were prepared using: 550 mg/kg of ascorbic acid (AA), 125 mg/kg of a starter culture (SC) (Bactoferm TRADI-302, Chr Hansen composed of *L. sakei*, *S. xylosus* and *S. carnosus* with proportions of 1:1:1 and formulated to reach an approximate concentration in the meat batter of 4 × 10^10^ cfu/g), 550 mg/kg of sodium ascorbate (SA) and 0.1% glucono-delta-lactone (GDL). Lean pork and fat were minced in a Mainca PC-98 mincer (Equipamientos Cárnicos S.L., Barcelona, Spain) with an 8 mm plate. Pork meat and fat were then mixed to a total of 30 kg and stored at 4 °C. This common batter with 16.9 ± 1.5% protein and 11.5 ± 1.9% fat was split into 5 batches and the formulas with different reducing agents were added and homogenized with a Talsa Mix25 homogenizer (Talsabell, S.A., Valencia, Spain).

Synthetic collagen casings (55 mm diameter) were filled using a Talsa H15 hydraulic piston-based sausage stuffer (Talsabell, S.A., Valencia, Spain) and 10–12 sausages of approx. 400 g were made to obtain a final product of approx. 260 g after the ripening process. Once the sausages were prepared, they were immersed at a depth of 1″ in a natamycin bath (1.2 g/L) to avoid the growth of mold on the surface of the casings.

Sausages were stored in warm conditions for 24 h (22 °C and a relative humidity of 95%) and then were dried at 11–12 °C for 29 days (to align with current regulations related to the use of nitrates as additives) in a ripening chamber from the Pilot Plant facilities of the University of Córdoba. The relative humidity was gradually reduced until a final value of 70%. After the ripening period, dry-fermented sausages presented less than the half of the upper content limit for this product as established by Spanish regulations [17]. Sausages were then vacuum-packed using an Original Henkelman Vacuum Systems packaging machine (Boxer 62, Henkelman, I Netherlands) and stored at 4 °C until the end of their shelf life (180 days). The process was repeated on two different dates in order to duplicate the experiment and guarantee reproducibility of the results.

### 2.2. Determination of Weight Loss, Moisture Content, Water Activity and pH

To study the effect of ascorbic acid, a starter culture, sodium ascorbate and glucono-delta-lactone on basic technological parameters of dry-fermented sausages, weight loss, moisture content, water activity and pH were determined. Weight loss was determined in 2 pieces per batch and replicated. Each sample was weighed during the ripening process (days 0, 3, 15 and 30) and at the end of its shelf life (30 + 180 days = day 210). Weight loss was calculated using the following arithmetical expression:(1)$$\%weight\;loss=\frac{\left(W_{0}-W_{d}\right)}{W_{0}}\times 100$$
where *W*_0_ is weight of the product on day 0 of processing (g) and *W_d_* is weight on the sampling day during the ripening process or the shelf life period (g). Moisture content was determined according to the AOAC [18]. Water activity (a_W_) was determined using a dew-point hygrometer (AquaLab 4, Decagon Device Inc., Pullman, WA, USA) according to the manufacturer’s instructions. To measure pH, the penetration probe of a pH-meter (model Edge, Hanna Instruments, USA) was introduced twice inside the mass of each sausage sample in triplicate. When the samples were dry (day 180), a knife was used to introduce the probe into the product. The pH electrode was recalibrated at room temperature every five samples, using two standard buffer solutions at pH 4.0 and 7.0, and rinsed between measurements.

### 2.3. Color Determination

Instrumental color was determined to evaluate the influence of the reductants added on the visual color of samples. The determinations were done in triplicate on the surface of a slice using a colorimeter (Konica Minolta CM-600d, Minolta Co., Osaka, Japan) with D65 standard illuminant, 8° visual angle and 8-mm measurement aperture, according to the L* (lightness), a* (redness) and b* (yellowness) system [19], and the average value was reported. The standardization of the instrument, with respect to the white calibration plate, was carried out before measurements were taken. The correction parameters of the CR-A43 white calibration plate were Y = 86.6, x = 0.3187 and y = 0.336.

### 2.4. Determination of Lipid Oxidation

In order to determine the effect of ascorbic acid, a starter culture, sodium ascorbate and glucono-delta-lactone on the antioxidant properties of nitrites, lipid oxidation was measured. The thiobarbituric acid reactive substances test (TBARS) described by Tarladgis et al. [20] was performed with minor modifications. The TBARS test was determined during the ripening process (days 0, 3, 15 and 30) and storage time (days 60, 90, 120, 150, 180 and 210) by two repetitions per replicate. Absorbance was measured at 532 nm in a UV/VIS spectrophotometer (Shimadzu UV-1700, Shimadzu Corporation, Kyoto, Japan). Values were expressed as mg of malonaldehyde per kg of product (mg MDA/kg of product).

### 2.5. Determination of Nitrate and Nitrite Content

To examine the capacity of the reducing agents used to reduce nitrates to nitrites, their concentrations were determined by following the spectrophotometric methods of BOE [21] and Higuero et al. [22] with minor modifications. Carrez I and II reagents were added to the homogenized sample. The ethanolic extract was prepared by diluting the previous solution with distilled water and ethanol while stirring and heating at 70 °C.

For nitrate measurement, brucine–sulfanilic acid reagent and sulfuric acid solution were added to a sample tube with the ethanolic extract. The tubes were then mixed with a scraper and incubated in darkness for 10 min. Before absorbance was read at 410 nm with a UV/VIS spectrophotometer (Shimadzu UV-1700, Shimadzu Corporation, Kyoto, Japan), the solution was diluted with distilled water. Nitrate results were calculated from a KNO_3_ standard curve using water as a blank and were expressed as mg NO_3_^−^/kg dry matter.

Nitrite was determined from the ethanolic extract by adding sulfamide and N-(1-Naphthyl) ethylenediamine dihydrochloride (NED) solutions. Mixtures were stirred with a scraper and incubated for 15 min at room temperature. The absorbance of samples was then read at 538 nm. Nitrite results were calculated from a NaNO_2_ standard curve (0.16–0.001 mM) and were expressed as mg NO_2_^−^/kg dry matter. All the determinations were performed in duplicate.

### 2.6. Statistical Analysis

Statistica 8.0 software (Stat Soft. Inc., Tulsa, OK, USA) for Windows was used to analyze the effects of the different reducing agents on evolution during the ripening process and shelf life period of physico-chemical parameters, instrumental color, nitrate and nitrite content and lipid oxidation. A two-way ANOVA was used to evaluate both effects, i.e., time and reducing agent. In order to compare the means of the different experiments in pairs, a Tukey’s honestly significant difference (HSD) post hoc test was used with *p* < 0.05.

## 3. Results

### 3.1. Physico-Chemical Determinations

Weight loss, moisture, a_W_ and pH values are presented in Table 1. During the ripening period, the samples lost a mean of 45.7% of their initial weight. Weight loss values reached a mean of 48.7% of initial weight by the end of the shelf life (day 210) of the vacuum-packed product stored at 4 °C. SC sausages lost less weight throughout the experiment (ripening and storage period) (*p* < 0.05) along with SA samples during the ripening process (*p* < 0.05) and AA samples by the end of the shelf life period (*p* < 0.05). GDL sausages presented the highest weight loss in both experimental periods (*p* < 0.05).

Moisture values diminished throughout the storage period of the products until average values of 38.7%. This decrease was particularly pronounced from day 15 to 30. Only at the end of shelf life were significant differences among the experiments observed with SC samples presenting the highest moisture values (*p* < 0.05).

As per moisture, a_W_ decreased throughout the ripening period until reaching average values of 0.878, and this trend was maintained during the storage period. The a_W_ was affected by the reducing agent on day 15 (*p* < 0.05), with C- samples showing the lowest values. As with moisture determination, SC sausages showed the highest a_W_ values by the end of the shelf life period (*p* < 0.05).

Regarding pH evolution, an accentuated decrease was observed from day 0 to day 15, with average values of 5.78 and 5.27 respectively. pH values were stable from day 15 to the end of the storage period. The effect of the reducing agent was particularly noticeable at the beginning of the experiment, where the glucono-delta-lactone samples presented pH values 0.10 points lower than the average. pH values for SC and AA sausages were among the highest on day 0; however, from day 3, the values dropped down in a more stable and gradual way.

### 3.2. Instrumental Color

As can be seen in Table 2, values for L* declined throughout the ripening period, from an initial mean value of 44.3 to an average value of 38.9 on day 30. L* was maintained during the storage period (39.9 on day 210). Significant differences were detected among sausages due to the reducing agent (*p* < 0.001). SC and SA sausages presented lower L* values throughout the whole experiment. On the contrary, AA samples presented the highest L*values.

Although no clear effect on a* was observed as a result of ripening and storage periods, AA, SC and SA samples showed growing values for this index, particularly during the first period (until day 30). No differences were found for a* due to the reducing agent; however, SC and SA samples presented the highest average values for the whole process (13.7 in both cases).

Results of b* were even more unclear than those of the previous parameter. The index rose in general from day 0 to 3, gradually dropped until the end of the ripening period (day 30) and finally rose during storage (day 210). The effect of the reducing agent was only significant (*p* < 0.05) on day 210 and SA sausages presented the highest values.

### 3.3. Nitrates and Nitrites

Mean values for nitrate content for the whole experiment ranged from 117.7 mg/kg in C- to 63.9 mg/kg in SC samples (Figure 1). Nitrate concentrations declined from the beginning until day 15–30; however, the drop was much more accentuated for AA, SC and SA sausages. Thereafter, the values remained stable with an average of 56.0 mg/kg until the end of the shelf life period. Nitrite concentrations rose from the beginning until the end of the experiment with values ranging from 0.3 mg/kg on day 0 to 2.7 mg/kg on day 210 (Figure 2). SC and GDL samples reached the highest values of residual nitrite with averages on day 210 of 4.4 mg/kg and 5.0 mg/kg, respectively.

### 3.4. Lipid Oxidation

The concentration of TBARS in the ripening and storage period is represented in Figure 3. Significant changes (*p* < 0.05) were observed in TBARS values among samples during storage. Values of TBARS stayed below 0.25 mg MDA/kg of sample throughout the ripening period. Thereafter, TBARS concentrations rose until day 150 and then dropped down. Particularly low TBARS values were observed in SC and SA samples throughout the storage period, with averages of 0.35 and 0.48 mg MDA/kg of sample, respectively. On days 60 and 150, differences between these two batches were particularly low and evident. On the other hand, GDL sausages showed TBARS mean values of 0.73 mg MDA/kg of sample during storage.

## 4. Discussion

### 4.1. Physico-Chemical Determinations

Weight loss due mainly to moisture loss is a consequence of progressive and partial dehydration of meat. Weight loss as observed in our study is in line with the loss described for long-ripened sausages [23]. However, despite this fact, the sausages from this study presented the usual profile of a short-ripened product: the moisture content at the end of the ripening period was around 35–40%, the moisture:protein ratio was above 1.5:1 and pH was below 5.3. The low fat content of the original batter could be behind this issue. Fat level affects the chemical composition and physico-chemical parameters of dry-fermented sausages [4]. These high water levels are typically described in dry-fermented sausages manufactured with low fat content [24]. Weight loss under vacuum conditions was maintained during the storage period in our samples, though to a lesser intensity than in the ripening phase. This phenomenon was in consonance with what has been reported previously for vacuum-packed low-fat dry-fermented sausages [25]. The reduction in fat content caused fast dehydration in sausages which in turn affected their technological parameters (a_W_, humidity and water loss) during ripening and storage periods. To amend this problem, a short ripening period (less than 1 month) is recommended for these sausages in order to avoid the economic costs that such great water loss entails in the final product while ensuring adequate a_W_ values guarantee the same hygienic conditions.

Regarding the effect of the reductant added, differences among batches were not relevant because even SC samples (those with lower weight loss and higher moisture percentage at the end of the storage period) presented a_W_ and pH values low enough to prevent any microbiological risk (below 0.92 and 5.3, respectively). These results are supported by other studies where low or not significant differences in water-related parameters were attributable to similar starter cultures used in the manufacture of sausages [26].

pH drop during fermentation of sausages is a consequence of lactic acid generation and accumulation due to lactic acid bacteria activity (which comes from autochthonous flora or from an added starter culture). pH decline controls the growth of pathogens and promotes protein coagulation [23]. Most samples from the experiment showed a slight and progressive pH drop which is typical for slow-fermented products. This mild pH decline contrasted with the pronounced reduction in pH for GDL samples at the beginning of the experiment due to its fast conversion into gluconic acid. The low pH values for GDL were not accompanied by a reduction in a_W_. Furthermore, this different trend with regard to the other batches was not maintained for pH during the whole experiment, and the significant differentiation of this parameter was only appreciable on day 0. Nevertheless, the denaturation of proteins in GDL samples because of the dramatic drop in pH was noticeable and was reflected by higher weight loss by the end of the ripening and storage period when compared to any other batch.

### 4.2. Instrumental Color

Luminosity values of the samples studied were low but in consonance with those described for low-fat sausages that have less fat (which is white) [25]. The decrease in L* values during ripening is a consequence of weight loss. Positive correlations were previously described between L* values and moisture and a_W_ in dry-fermented sausages [27], and samples became darker during the dehydration that takes place in the ripening period.

The reduction of nitrates to nitrites, followed by the production of nitric oxide, is a key point in developing the characteristic reddish color of fermented sausages. Nitric oxide reacts with myoglobin, producing nitrosylmyoglobin that gives the attractive appearance to dry-fermented meat products [28]. The increase in a* during fermentation was evident for AA, SC and SA samples when compared to C-. The reaction between myoglobin and nitric oxide was probably more intense and enhanced by low pH values and by the action of these added reductants [25]. Although differences between day 30 and 210 were not significant, it seems that the red color of SC samples was the most stable. For SA samples, a* values were even higher by the end of the shelf life of the product. Sodium ascorbate and the starter culture exerted an antioxidant effect that prevented the discoloration of the product in a more effective manner. The improvement of some quality traits. such as the desirable cured red color due to the use of combinations of nitrite and ascorbate. has been previously described [29]. The fast pH decline at the beginning of the fermentation step for GDL samples had a negative impact on sausage color due to the inhibition of spontaneous fermentations mediated by native strains of coagulase negative staphylococci (CNS) [23]. As a result, the reddish color development was much more inconsistent in this last batch.

No clear trend was observed for b* values. Yellowness was not clearly affected by time or the reducing agent. Both a* and b* values were low compared to the previous literature [30]. Low fat content and the absence of pigmented colorants and species (such as paprika with high carotenoid content) sensitive to oxidation processes might be the reason.

### 4.3. Nitrates and Nitrites

Nitrate and nitrite content was consistent with that observed by other authors in dry-cured fermented sausages [31], chorizo [30] or dry cured loins [22] using a similar initial formulation. Reduction reactions in samples with added nitrate mainly depend on the nitrate reductase activity of staphylococci but also may be mediated by reducing agents. In fact, SC (with CNS among the microorganisms added), but also SA and to a lesser extent AA batches, was the most effective nitrate reductant in this study. Indeed, the addition of sodium ascorbate prevents nitrosation reactions in meat-cured products [31].

Nitrite can be partially oxidized to nitrate by sequestering oxygen [28]. Hospital et al. [9] reported increases in nitrate levels during the drying process of dry-fermented sausages as the highest final residual values of the whole ripening period. This extreme was not observed in C- and GDL samples, but nitrate levels above 100 mg/kg were observed in both batches by the end of the storage period. However, as the high levels observed in C- might be due to a lack of reducing agents in the initial batter, the high levels detected in GDL samples were probably related to the inhibition of CNS because of the fast pH decline at the beginning of the ripening period.

### 4.4. Lipid Oxidation

TBARS values observed during the ripening period were lower than those reported by other authors for similar products, such as chorizo [30] or venison salchichón [32]. Low TBARS values are expected in samples with low fat content [3]. However, our results were in line with the TBARS values observed by Ozaki et al. during the storage period in regular dry sausages [5]. The typical increase in the concentration of TBARS was observed but late, coming at the end of the shelf life of samples. The antioxidant effect of nitrites formed from added potassium nitrate and the low fat content of the sausages analyzed could be the reason for the delay that samples experienced in relation to appearance of the oxidation peak. The final drop in TBARS values was also observed previously and was attributed to the instability of malonaldehyde [27].

TBARS results highlighted the superoxide dismutase and catalase activities of CNS when included in the starter culture used in SC samples. These enzymes limit oxidative processes by neutralizing pro-oxidant molecules [33]. In addition, the antioxidant effect of the sodium ascorbate on SA sausages was also noticeable when comparing results of TBARS with those of C-, AA or GDL batches. This is in agreement with Berardo et al. [34] but contrary to the results reported by Bonifacie et al. [31], who described no antioxidant effect for sodium ascorbate when it was added in combination with sodium nitrite. When AA is added to a cured meat product it accelerates all the reducing steps, such as the formation of nitric oxide from nitrite which in turn can result in low residual nitrite levels in the product [35]. Sánchez-Escalante [36] also reported that the antioxidant activity of AA might depend on dosage. In fact, in some conditions, ascorbic acid can even act as a pro-oxidant. All these issues might be behind the differences between the antioxidant activities of AA and SA samples. Regarding the GDL batch, although a low pH environment, promoted by glucono-delta-lactone, might have enhanced nitrite reactivity [8] and its antioxidant properties, the fast pH decline probably inhibited CNS. These microorganisms likely make a crucial contribution to preventing lipid oxidation. Apart from these differences between batches, in all cases the values were well below the threshold of 2 mg/kg of TBARS that indicates meat rancidity [37].

## 5. Conclusions

This study provides important information about the use of different reductants and the evolution of technological parameters during the whole shelf life period of this low fat content product.

High nitrate values were observed throughout the entire experiment. This fact emphasized the necessity of modifying the strategy used for 1-month-ripened sausages. Two alternatives might be considered instead of exclusively adding nitrate: use of a combination of nitrate and nitrite or use of only nitrite with sausages ripened for less than 1 month. Starter culture and sodium ascorbate have shown the best reducing and antioxidant activities among the additives studied.

Finally, once we have this sausage with low fat content technologically defined, the next step will be identifying the sensory characteristics of the product in order to successfully design a novel offer to the market that meets consumer needs.

## Figures and Tables

**Figure 1 foods-11-02606-f001:**
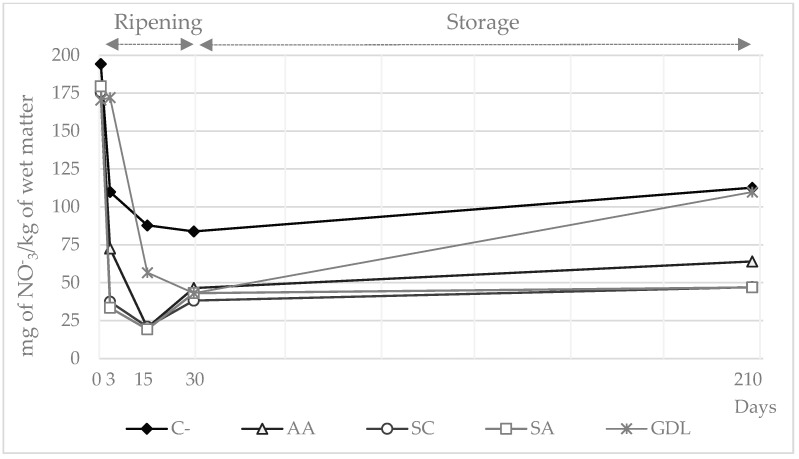
Nitrate concentration (mg/kg of wet matter) in sausages manufactured with different reducing agents added during ripening and storage periods (days). C- = negative control; AA = ascorbic acid; SC = starter culture; SA = sodium ascorbate; GDL = glucono-delta-lactone.

**Figure 2 foods-11-02606-f002:**
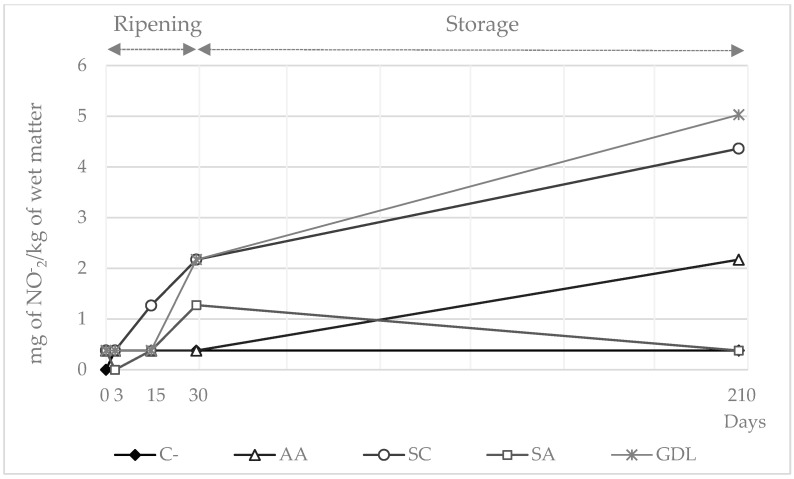
Nitrite concentration (mg/kg of wet matter) in sausages manufactured with different reducing agents added during ripening and storage periods (days). C- = negative control; AA = ascorbic acid; SC = starter culture; SA = sodium ascorbate; GDL = glucono-delta-lactone.

**Figure 3 foods-11-02606-f003:**
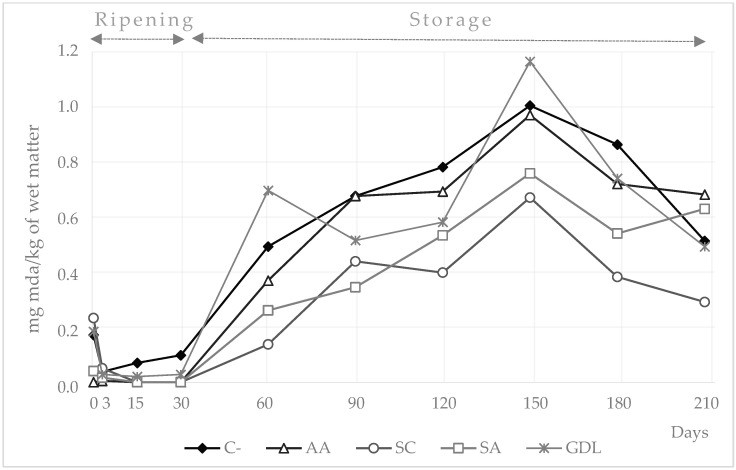
TBARS values (mg MDA/kg of wet matter) in sausages manufactured with different reducing agents added during ripening and storage periods (days). C- = negative control; AA = ascorbic acid; SC = starter culture; SA = sodium ascorbate; GDL = glucono-delta-lactone.

**Table 1 foods-11-02606-t001:** Means and standard deviations for weight loss and physico-chemical parameters of sausages manufactured with different reducing agents added during ripening and storage periods.

	Day 0	Day 3	Day 15	Day 30	Day 210	D	E	D × E
	Weight loss (%)							
C-	-	15.70 ± 0.10 ^aA^	34.55 ± 0.55 ^b^	46.04 ± 0.30 ^c^	49.20 ± 0.21 ^dAB^	***	***	*
AA	-	15.05 ± 0.05 ^aAB^	33.15 ± 0.74 ^b^	45.77 ± 0.15 ^c^	48.00 ± 0.53 ^cB^			
SC	-	15.35 ± 0.15 ^aA^	33.40 ± 0.70 ^b^	44.65 ± 0.25 ^c^	48.24 ± 0.09 ^dB^			
SA	-	14.00 ± 0.10 ^aB^	33.85 ± 0.65 ^b^	45.75 ± 0.35 ^c^	50.03 ± 0.04 ^dA^			
GDL	-	15.55 ± 0.45 ^aA^	34.75 ± 0.33 ^b^	46.45 ± 0.62 ^c^	50.17 ± 0.06 ^dA^			
	Moisture (%)							
C-	66.46 ± 1.08 ^a^	67.71 ± 0.26 ^a^	60.76 ± 0.02 ^b^	40.07 ± 1.57 ^c^	39.20 ± 0.02 ^cBC^	***	ns	*
AA	69.15 ± 2.28 ^a^	66.08 ± 0.07 ^a^	62.27 ± 0.48 ^a^	36.77 ± 3.97 ^b^	39.03 ± 0.05 ^bC^			
SC	67.80 ± 0.14 ^a^	63.73 ± 1.06 ^ab^	61.12 ± 1.16 ^b^	40.74 ± 2.61 ^c^	42.64 ± 0.68 ^cA^			
SA	69.67 ± 0.02 ^a^	66.51 ± 0.41 ^a^	60.99 ± 1.88 ^b^	38.66 ± 0.41 ^c^	41.30 ± 0.65 ^cAB^			
GDL	68.60 ± 1.10 ^a^	65.60 ± 2.50 ^ab^	61.50 ± 0.90 ^b^	36.90 ± 0.20 ^c^	40.55 ± 0.80 ^cABC^			
	a_W_							
C-	0.967 ± 0.001 ^a^	0.970 ± 0.005 ^a^	0.929 ± 0.001 ^bB^	0.876 ± 0.010 ^b^	0.833 ± 0.001 ^bB^	***	***	ns
AA	0.975 ± 0.001 ^a^	0.974 ± 0.001 ^a^	0.945 ± 0.004 ^bA^	0.889 ± 0.003 ^b^	0.838 ± 0.096 ^bB^			
SC	0.978 ± 0.003 ^a^	0.972 ± 0.010 ^a^	0.937 ± 0.004 ^bAB^	0.879 ± 0.001 ^b^	0.846 ± 0.044 ^bA^			
SA	0.972 ± 0.009 ^a^	0.974 ± 0.004 ^a^	0.924 ± 0.006 ^bB^	0.875 ± 0.001 ^b^	0.835 ± 0.005 ^bB^			
GDL	0.975 ± 0.002 ^a^	0.972 ± 0.007 ^a^	0.934 ± 0.002 ^bAB^	0.872 ± 0.002 ^b^	0.834 ± 0.007 ^bB^			
	pH							
C-	5.84 ± 0.01 ^aA^	5.38 ± 0.07 ^b^	5.30 ± 0.01 ^bcAB^	5.28 ± 0.01 ^bc^	5.20 ± 0.06 ^c^	***	ns	***
AA	5.83 ± 0.02 ^aAB^	5.33 ± 0.03 ^b^	5.24 ± 0.01 ^bcBC^	5.23 ± 0.04 ^bc^	5.20 ± 0.03 ^c^			
SC	5.81 ± 0.01 ^aAB^	5.38 ± 0.01 ^b^	5.33 ± 0.02 ^cA^	5.26 ± 0.01 ^d^	5.17 ± 0.01 ^e^			
SA	5.76 ± 0.03 ^aB^	5.37 ± 0.03 ^b^	5.22 ± 0.02 ^cC^	5.25 ± 0.01 ^c^	5.21 ± 0.01 ^c^			
GDL	5.68 ± 0.01 ^aC^	5.43 ± 0.05 ^b^	5.27 ± 0.01 ^cABC^	5.25 ± 0.01 ^c^	5.28 ± 0.01 ^c^			

ns: non-significant; * *p* < 0.05; *** *p* < 0.001. ^a–e^ Mean values in the same row with different superscript presented significant differences (*p* < 0.05). ^A–C^ Mean values in the same column with different superscript presented significant differences (*p* < 0.05). D = day; E = experimental formula; D × E = day × experimental formula interaction; C- = negative control; AA = ascorbic acid; SC = starter culture; SA = sodium ascorbate; GDL = glucono-delta-lactone.

**Table 2 foods-11-02606-t002:** Means and standard deviations of instrumental color for sausages manufactured with different reducing agents added during ripening and storage periods.

	Day 0	Day 3	Day 15	Day 30	Day 210	D	E	D×E
	Lightness (L*)							
C-	47.6 ± 3.0 ^aA^	38.8 ± 3.2 ^bB^	41.1 ± 1.2 ^bAB^	39.0 ± 2.2 ^bAB^	39.0 ± 3.6 ^bBC^	***	***	***
AA	45.2 ± 3.1 ^aAB^	43.6 ± 3.2 ^abA^	42.0 ± 2.4 ^abA^	40.6 ± 1.8 ^bA^	45.7 ± 2.4 ^aA^			
SC	39.9 ± 1.9 ^abC^	42.3 ± 2.4 ^aAB^	38.6 ± 1.3 ^bC^	38.1 ± 2.6 ^bAB^	37.7 ± 3.8 ^bC^			
SA	42.1 ± 2.5 ^abBC^	44.8 ± 1.7 ^aA^	40.3 ± 1.5 ^bABC^	39.3 ± 3.1 ^bcAB^	36.3 ± 2.0 ^cC^			
GDL	46.7 ± 3.6 ^aA^	42.8 ± 2.0 ^bA^	39.3 ± 2.0 ^cBC^	37.5 ± 1.5 ^cB^	41.2 ± 2.7 ^bcABC^			
	Redness (a*)							
C-	10.91 ± 1.60	13.19 ± 0.30	13.90 ± 1.52	15.31 ±1.70	13.00 ± 0.28	***	ns	ns
AA	9.20 ± 0.10 ^b^	14.26 ± 0.30 ^a^	13.98 ± 1.50 ^a^	15.47 ±0.90 ^a^	12.94 ± 0.40 ^a^			
SC	9.84 ± 0.25 ^b^	14.07 ± 1.74 ^ab^	14.16 ± 1.01 ^ab^	15.47 ±2.21 ^a^	14.94 ± 0.71 ^ab^			
SA	10.25 ± 0.86 ^b^	13.92 ± 0.60 ^a^	14.10 ± 0.84 ^a^	13.69 ±0.89 ^a^	16.51 ± 0.19 ^a^			
GDL	8.46 ± 0.22	13.53 ± 2.23	15.63 ± 1.78	14.15 ± 1.90	13.61 ± 2.24			
	Yellowness (b*)							
C-	8.46 ± 0.09	7.91 ± 0.42	7.23 ± 2.12	7.73 ± 0.38	8.10 ± 0.59 ^B^	***	ns	ns
AA	6.99 ± 0.30	9.19 ± 0.10	7.37 ± 1.30	8.19 ± 0.70	9.25 ± 0.10 ^AB^			
SC	6.38 ± 0.44	9.21 ± 0.74	7.79 ± 0.75	7.34 ± 1.50	8.55 ± 0.23 ^B^			
SA	7.44 ± 0.46 ^b^	9.26 ± 1.05 ^a^	7.30 ± 0.01 ^b^	7.17 ± 0.25 ^b^	9.87 ± 0.10 ^aA^			
GDL	6.14 ± 1.15	8.78 ± 0.64	8.73 ± 1.37	6.83 ± 1.60	8.20 ± 0.17 ^B^			

ns: non-significant; *** *p* < 0.001. ^a–c^ Mean values in the same row with different superscript presented significant differences (*p* < 0.05). ^A–C^ Mean values in the same column with different superscript presented significant differences (*p* < 0.05). D = day; E = experimental formula; D × E = day × experimental formula interaction; C- = negative control; AA = ascorbic acid; SC = starter culture; SA = sodium ascorbate; GDL = glucono-delta-lactone.

## Data Availability

Data are contained within the article.

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
