# Peer review of "Influence of Common Reducing Agents on Technological Parameters of Dry-Fermented Sausages with Low Fat Content"

_foods, 2022, doi:10.3390/foods11172606_

Round 1
Reviewer 1 Report
The results obtained for TBARS, Nitrite and Nitrate need to be tabulated, as the graphs presented are difficult to visualize and understand
Author Response
Dear referee,
We appreciate the time and effort that you have dedicated to providing your valuable feedback on our manuscript. We have now consider the suggestions provided and we think that the results of our research are much more interpretable and better understandable for the reader in visual format. We have tried to improve the graphs to clearly differentiate the evolution of each batch as you can see in the new version of our manuscript.
Thank you again for your time, we hope our modifications meet your approval.
The authors
Reviewer 2 Report
Abstract:
Please give some perspectives and applications of your research.
Keywords:
Please avoid to use those words already present in the title.
Introduction:
I suggest adding something about the benefits of low-fat dry fermented sausage. The influence of reducing agent on dry fermented sausage is not seen in this introduction.
Materials and Methods:
2.1: How much fat is added?
Line 68-69: 125 ppm of a starter culture (SC) (Bactoferm TRADI-302, Chr Hansen composed by L. sakei, S. xylosus and S. carnosus). What is the ratio of three strains? What is the vaccination amount?
Line 98-99: Remove the comma in the formula. Please check the formula.
Line 113-114: “The standardization of the instrument with respect to the white calibration plate was carried out before measurements were taken.” What are the white calibration plate correction parameters?
In "material and methods" part, a brief explanation of the aims of each methodology is needed.
What are the microbiological and sensory indicators of sausage during storage?
Proximate basic composition and energy value?
Conclusions:
Please add conclusion to stress out the novelty and future perspectives of the work.
References:
I suggest avoiding lumping the references. Each reference should be discussed separately or deleted older.
What is the low fat standard of low fat dry fermented sausage?
Author Response
Dear referee,
We appreciate the time and effort that you have dedicated to providing your valuable feedback on our manuscript. We are grateful your comments on our paper. We have been able to incorporate changes to reflect of the suggestions provided. We have highlighted the changes within the manuscript. You can see the answers and comments to your suggestions in the document attached.
The authors

Reviewer 3 Report
The article is about investigation of few additives during fermentation of dry sausages. The article needs revision.
1. The language of the article needs sympathetic consideration.
2. Line 15: Write ‘reducing agents’
3. L27: Rephrase
4. L28: Delete ‘valued and demanded’
5. L29-30: Rephrase
6. L33: Delete ‘kind of’
7. L33-36: Remove ‘Nevertheless’
8. L47-48: Rephrase
9. L51-53: Avoid incomplete sentences
10. Table 1 needs to be readjusted. The basic composition is the same and hence should not be repeated. Just mention the variable ingredients.
11. L180: State full form of the abbreviation used
12. Table 2 and 3 and in figures: revise captions. Don’t use the word ‘evolution’. Describe all the abbreviations used in tables and figures
13. L216: State full form
14. L218: Is it an SI unit? Give full form
15. L251: The addition of any starter culture is not mentioned in the methods or in the results. Provide details about the production of starter culture and its charactersitcs
Author Response

(The authors gave the same response as above.)

Round 2
Reviewer 2 Report
The author has modified it. Thanks!
Author Response
Thank you for your time and your valuable comments and suggestions.
Kind regards,
The authors